# Evaluating Indigenous Identity and Stress as Potential Barriers to Accessing On-Campus Healthcare for Indigenous Students at a Large Southwestern University

**DOI:** 10.3390/ijerph22091409

**Published:** 2025-09-10

**Authors:** William O. Carson, Michelle Valenti, Kendrick Begay, Scott Carvajal, Stephanie Russo Carroll, Nicole P. Yuan, Felina M. Cordova-Marks

**Affiliations:** 1Mel and Enid Zuckerman College of Public Health, University of Arizona, Tucson, AZ 85721, USA; michellevalenti@arizona.edu (M.V.); carvajal@arizona.edu (S.C.); stephaniecarroll@arizona.edu (S.R.C.); nyuan@arizona.edu (N.P.Y.); felina@arizona.edu (F.M.C.-M.); 2Native Nations Institute, Udall Center for Studies in Public Policy, University of Arizona, Tucson, AZ 85721, USA; 3College of Engineering, University of Arizona, Tucson, AZ 85721, USA; kbegay69@arizona.edu

**Keywords:** Indigenous identity, perceived stress, colleges and universities, healthcare access, healthcare utilization

## Abstract

Introduction: This study examined the relationship between Indigenous identity, perceived stress, and healthcare utilization for Indigenous students on-campus. Methods: Potential participants included undergraduate and graduate Indigenous students from Tribal Nations within the United States. Participants were recruited through community partnerships and in person communication. This survey included the Perceived Stress Scale (PSS-10), the Multiethnic Identity Measure (MEIM), and university Campus Health Service Health and Wellness Survey. Results: 153 Indigenous students from United States-based Tribal Nations participated in this survey. While there appears to be a relationship between Indigenous identity, perceived stress, and Campus Health utilization, the results were not significant for the second tertile (OR: 1.1 (0.4, 2.7)) or third tertile (1.4 (0.5, 3.3)). Students who reported “Yes” or “Unsure” to questions on if their insurance needed them to go outside of the university were far less likely to use Campus Health (Yes OR: 0.2 (0.08–0.5)); (Unsure OR: 0.09 (0.03–0.3)) and CAPS (Yes OR: 0.2 (0.09–0.6)); (Unsure OR: 0.2 (0.04–0.4)). Discussion: This study saw a complex relationship between Indigenous identity, perceived stress, and campus health utilization; however, the findings are not statistically significant. There are distinctions in on campus health care usage when adjusting for undergraduate or graduate student status and health insurance literacy. Conclusion: The research findings offer many promising avenues for future work around Indigenous identity, affordability of healthcare, and importance of health literacy.

## 1. Introduction

At the large Southwestern University which this study takes place at, there are more than 2000 Indigenous students with nearly 450 being enrolled graduate students [1]. Recent studies have examined the great benefit to tribal nations that college graduates have and the need for universities to better support Indigenous students through their time at school [2]. While the university is achieving success in getting students enrolled, there remain challenges in retention and student utilization rates of certain University services, notably accessing healthcare programs. Indigenous students have the lowest reported healthcare utilization rates on campus [3] with no clear understanding as to why. The university has a variety of services that students can access to increase their connection to the university Indigenous community, such as Native American Student Affairs (NASA), the Indigenous cultural center on campus, assortments of clubs for various interests, and degree programs and courses that focus on an Indigenous curriculum and worldview. Despite the considerable number of services available to Indigenous students, there remains additional barriers.

Numerous studies have explored the relationship between identity and stress among Indigenous students on college campuses [4,5,6,7,8,9]. However, the findings regarding the impact of a strong Indigenous identity have been inconsistent. A review of the literature reveals a lack of consensus on how Indigenous identity influences stress levels [10]. For instance, one investigator examined changes in Indigenous students’ identities throughout their college careers and found minimal change in Indigenous identity [6]. Conversely, a different research team identified a strong Indigenous identity as a protective factor when examining the impact of historical loss on psychological distress [7]. Lastly, in another recent study with Alaska Native students, it was demonstrated that elder-led programs aimed at strengthening community ties and enhancing Indigenous identity yielded positive outcomes. Researchers in this study noted improved scores for students’ cultural identity [4]. While identity has been examined in studies with Indigenous college students, there is more limited research on the impact Indigenous identity may have on accessing services and resources provided while at school.

Determining healthcare utilization for college students is challenging due to limited details, as their status as students and living situation often differ from their home address or the place they identify as their origin. A recent study of on-campus health care providers documented that 60% of healthcare visits by students are for primary care and 13% for mental health services [11]. Despite the novel findings of this study, the authors note that more work needs to be done to better understand the differences in on-campus healthcare utilization for different racial and ethnic groups [11]. It has been noted that even with similar insurance levels and access to care, BIPOC student populations are less likely to utilize health services than white students [12]. This limited amount of research is further exacerbated when attempting to only examine healthcare utilization by Indigenous students, as even for the general Indigenous population, healthcare utilization research is sparse, with the most recent data coming from the Centers for Disease Control and Prevention [13].

Currently, the literature is focused on forms of social support for Indigenous college students on campus, with less attention toward challenges with health services, such as with the on-campus healthcare provider. As the Indigenous student population continues to grow, Campus Health (CH), the university’s healthcare provider for students, continues to see a decline in Indigenous students that utilize their medical and psychological services. Campus Health provides access to primary care services, urgent care, lab testing, x-ray services, physical therapy and referrals for students, staff, and faculty at the university. In addition, CH also has a psychological service unit, Counseling and Psych Services (CAPS), which provides scheduled counseling services, crisis based, and site-based counseling with a set co-pay. In addition, CAPS offers additional services across the university at Cultural Resource Centers (CRC) such as NASA at no cost, with embedded CAPS counselors for each center. As of July, 2025, the university consolidated the CRCs and relieved the directors of duty, however, the embedded counselors remain for the time being [14].

The university has one of the largest populations of Indigenous students at a higher education institution in the country. Despite this, there is limited to no research conducted with Indigenous university students that examines the relationship between Indigenous identity, stress, and healthcare utilization. Currently, Indigenous students have the lowest rates of utilization of on campus health care [3]. Prior research with BIPOC populations has explored the relationship between a strong ethnic identity and healthcare utilization [15,16]. University staff with NASA and Campus Health, as well as faculty serving as advisors to Indigenous students, hypothesized that Indigenous identity may be playing a role in utilization of health services, with a held assumption being that students with stronger Indigenous identities will be less likely to access on-campus healthcare services. This community-based research study set out to test this hypothesis in collaboration with Indigenous serving organizations across the university.

## 2. Materials and Methods

### 2.1. Community Partnerships and Approvals

This study emerged out of discussions with Campus Health, the university’s on campus healthcare provider, available to undergraduate and graduate students, faculty, and staff, and NASA prior to submitting grants for the proposed project. Letters of support were gathered, and all partners were engaged from pre-submission through dissemination of results. This study partnered with both organizations to identify potential issues with how and why Indigenous students do not use on campus healthcare and through these relationships, provided feedback on how the survey would be designed, including using CH materials. As part of the process in creating this research project, letters of support were acquired from Campus Health, NASA, and Native American Advancement and Tribal Engagement (NAATE). As part of the agreement to support the project, all three community partners would be engaged throughout the research process and would be given final report outs at the conclusion of the study and analysis was complete.

This study received IRB approval from the university, and all participants gave informed consent, with the consent document needing to be read and signed at the beginning of the survey.

### 2.2. Inclusion/Exclusion Criteria

The study aimed to be highly inclusive by involving individuals over 18 years of age who were enrolled as undergraduate or graduate students at the university and self-identified as part of an Indigenous People within the United States. All Tribal Nations within the United States, including federally recognized, state recognized, and unrecognized, were eligible. The only significant exclusion criteria were for self-identifying Indigenous people who were either not college students or identified with an Indigenous People outside the United States.

### 2.3. Survey Development 

#### 2.3.1. Exposure Measure

Indigenous identity was ascertained using the Multigroup Ethnic Identity Measure (MEIM). The MEIM is a brief survey used to assess an individual’s relationship with their self-identified ethnic group [17]. The MEIM has been tested for validity across ethnic and racial groups [18]. The MEIM has been utilized to measure ethnic identities’ status as a buffer against stress in minority college student populations in other studies, showing promise to be utilized with Indigenous students as well [5,6].

#### 2.3.2. Outcome Measure

Outcome variables were defined from utilizing the Healthcare Utilization questions from the university CH Health & Wellness Survey question bank, with express permission from our community partner, CH. This survey instrument is internally validated, and data can be found readily on the CH website [19]. Campus healthcare utilization was defined as self-reported use of either campus health medical services or counseling & psych services (CAPS). Participants indicated whether they had used the services (“Have you used any of the following services?”) responding with yes or no. We evaluated medical use (yes vs. no) and CAPS use (yes vs. no).

#### 2.3.3. Covariates 

The Perceived Stress Scale-10 (PSS-10) is a 10-question survey that measures the self-reported stress levels that an individual has felt over the past month [20]. The PSS-10 has been shown to be reliable with a multitude of populations, including Indigenous people [18]. To calculate PSS-10, the two subscales are totaled and then divided into three categories based on the score; cumulative scores below 13 are rated as “low”, scores 13 or greater and 26 or less are rated as “moderate”, and scores above 26 are rated as “severe”.

The PSS-10 has been utilized in Native populations to determine stress levels in AI/AN caregivers with their effect on health status and resilience [21,22]. We identified covariates with a priori based on literature, including gender, University class standing, undergraduate or graduate, and self-identified racial classification. Due to small sample sizes, we categorized gender as male or female, class standing as graduate or undergraduate, and race as Indigenous or multiracial.

#### 2.3.4. Modification of Instruments

We received our demographic questions from CH in addition to the healthcare utilization survey. These were not modified as suggestions for improvement based on usage were part of the deliverables to CH. There were several questions added to the demographic section that are Indigenous specific. These questions asked about participants’ Tribal Nations’ census region and enrollment status with their Tribal Nation. Consideration was given as to how detailed of data we would request for tribal affiliation and after consultation with the rest of the research team and adhering to the principles of Indigenous Data Sovereignty, data was limited to very wide range census regions so no Tribal Nation specific data can be determined or need to be reported out on. The MEIM, as originally designed, asks participants about their “ethnic group.” While this may be effective for certain populations, this would lead to potential surprises or rebuffing of the contents of the study if kept in. Therefore, phrasing was changed to “Tribe/Tribal Nation(s)” to better reflect our Indigenous identifying population.

#### 2.3.5. Data Collection & Study Population

We collected information via the Service Utilization and Media question bank from the university Campus Health Service Health and Wellness Survey. Campus Health uses this validated measurement tool annually for their own evaluation. We identified survey questions from the question bank that included campus health usage and questions to understand where else Indigenous students may go for healthcare. Participants completed the surveys using an online survey administration tool (REDCap) via smartphone, computer, or another device.

Study data were collected and managed using REDCap electronic data capture tools [23,24]. REDCap (Research Electronic Data Capture) is a secure, web-based software platform designed to support data capture for research studies, providing (1) an intuitive interface for validated data capture; (2) audit trails for tracking data manipulation and export procedures; (3) automated export procedures for seamless data downloads to common statistical packages; and (4) procedures for data integration and interoperability with external sources.

Recruitment of participants took place from October 2024 to February 2025. Recruitment was conducted through opportunistic sampling, fliers at cultural centers, promotion on university email list servs, and tabling at events such as the university’s Indigenous Peoples Day celebration. The study also employed the use of a paid undergraduate assistant whose primary job was to recruit additional students in demographic groups that were harder to reach, namely male identifying and undergraduate student populations. To improve participation, this study offered an incentive of a $15 e-gift card to a store of the participant’s choosing at the completion and verification of the survey. In total, 225 students completed the questionnaire with 153 submitting the completed documentation. The remaining students either started the survey and did not enter any answers or halted their progress in the demographic section.

### 2.4. Statistical Analysis

#### 2.4.1. Power Analysis

Analysis for calculating sample size was conducted in OpenEpi, Version 3 [25]. The population size for this was pulled from the demographic section of the NAATE website [1]. Our hypothesized percentage frequency of outcome factors in the population was set to 50% ± 5 and our confidence limits were set to 5%. For a confidence level of 80%, our calculation estimates we need a sample size of 152. As we reached a sample of 153, we achieved power for the study. The equation is listed here. Sample size n = [DEFF × Np(1 − p)]/[d^2^/Z^2^_1−a/2_ × (N − 1) + p × (1 − p).

#### 2.4.2. Primary Analysis

Both primary and secondary analyses were performed in SAS 9.4. [26]. First, descriptive statistics were calculated. We described categorical variables as the N, n, and proportion, while we used the mean and standard deviation (SD) for continuous variables. We conducted a stratified analysis, stratified descriptive statistics by the total population and the MEIM tertiles, which has been previously done with Indigenous populations [27]. To evaluate the association between the MEIM score (continuous) and campus health medical service use, we conducted logistic regression models. We adjusted the models for the score on the PSS-10, graduate student status, gender identification, and race.

#### 2.4.3. Confounders

Because affordability can affect health care usage, we conducted a series of tests for confounders to evaluate their effects. Participants indicated whether they had not accessed medical services when sick or mental health services due to affordability (“Since August 2023, have you needed any of the following but did not seek help because you couldn’t afford it?”). We conducted an additional sensitivity analysis excluding individuals who reported that their insurance required them to seek healthcare elsewhere (“Do you have medical insurance that requires you to go somewhere other than Campus Health for medical care?”). The Campus Health healthcare utilization survey does not mention Indian Health Service (IHS) either as an alternative provider or insurance. As a result, we used the above question as a proxy for students who believe their primary health insurance requires them to go to specific locations for care. IHS is not health insurance but if one uses the facilities as the primary means of healthcare, they will need to go back to their main facility for most care.

## 3. Results

A bot attack initially caused the study to receive thousands of fake responses and hundreds of messages to the PI’s inbox. With the assistance of the university REDCap administrators, we created a monitoring variable to ensure all responses met two specific criteria for analysis and compensation. The variable checked if survey takers selected “Native American/Alaska Native” in the demographics section. Almost all bot attacks did not select this option. We automatically deleted all participants who did not select “Native American/Alaska Native.” Additionally, to receive compensation, individuals had clear instructions to enter their university-affiliated email addresses.

Initially, prior to screening, we observed a total of 225 students who started the survey. After screening to include only those who completed the question on “Native American/Alaska Native,” we were left with 186 participants. The rest were not removed in the initial screening as they did not respond to any survey questions. We excluded individuals with missing MEIM information (n = 30) and those with missing campus health utilization information (n = 3) utilizing list wide deletion. The final analysis included 153 Indigenous students, as shown in Figure 1.

### 3.1. Demographics

Demographic information was collected and subsequently summarized in Table 1. Tertiles were utilized in this study to potentially see the distribution of MEIM scores as opposed to just measuring continuously as the MEIM does not have a standard cut-off or categories. Separating out allowed us to examine the scores in thirds with reasonable distributions and sample sizes. MEIM Tertile 1 had a sample size of fifty-two with scores ranging from 2.1–4.1. MEIM Tertile 2 had a sample size of forty-six with scores ranging from 4.2–4.6. Finally, MEIM Tertile 3 had a sample size of fifty-five with scores ranging from 4.7–5.0. We observed that 77.1% of participants identified as female, and 20.3% identified as male. Additionally, 4.6% identified as another gender identity, but due to small numbers, they will not be reported here. In our sample, 69.9% of participants identified as undergraduates while the remaining 29.4% identified as graduate students. One participant did not select either category, which was then coded as missing.

Students selected all racial and ethnic groups they identified as, with 27.5 percent identified with a racial/ethnic identity in addition to “Native American/Alaska Native.” Students self-reported their enrollment status in their Tribal Nations, which did not impact their standing for the survey. We observed that 92.8 percent of participants are enrolled in their Tribal Nations, while 6.5 percent reported not being enrolled. The last question asked students if they are affiliated with more than one Tribal Nation, and 16.3 percent said they are affiliated with more than one.

### 3.2. Healthcare Utilization and Insurance

In this selected sample, we aimed to determine how many participants utilized Campus Health services. Forty-nine students (31.1%) reported using Campus Health, while only forty-two (26.8%) had accessed CAPS for psychological services. This number contrasts significantly with the number of students who considered using CH and/or CAPS; ninety-four students (61.4%) indicated they would consider using on-campus healthcare services.

Health insurance coverage may have an impact on students’ healthcare options. We asked participants multiple questions about their health insurance and whether their insurance required them to go to specific locations. The results varied. The highest percentage (43.8%, n = 67) reported having an undisclosed plan on their parents’ health insurance, with an additional group specifically referencing ACCHS (16.3%, n = 25), Arizona’s version of Medicaid. Another group reported having a university-sponsored plan (14.4%, n = 22). A significant percentage of participants reported either having a different type of health insurance plan (14.4%, n = 22) or no health insurance at all (10.5%, n = 16).

We also asked whether participants knew if their health insurance required them to go to facilities outside of Campus Health. The responses were nearly evenly split: fifty students (32.7%) said their insurance did not require them to go elsewhere, forty-eight (31.4%) said they did have to go to other facilities for care, and fifty-four (35.3%) were unsure if their insurance had such requirements. There was also one missing variable for this section. The results can be seen in Table 2.

In our study, individuals who believed their insurance required them to go off-campus for healthcare were 80% less likely to use their insurance for Campus Health services (OR = 0.2, 95% CI: 0.08–0.5), and those unsure about this requirement were 91% less likely (OR = 0.09, 95% CI: 0.03–0.3). Similarly, for CAPS, individuals whose insurance required them to go elsewhere were 80% less likely to use their insurance (OR = 0.2, 95% CI: 0.09–0.5), and those unsure were also 80% less likely (OR = 0.2, 95% CI: 0.04–0.4). These findings suggest that both certainty and uncertainty about insurance requirements significantly reduce the likelihood of using insurance for Campus Health and CAPS services. This can be seen in Table 3.

### 3.3. Healthcare Utilization and MEIM Score

The next part of the study was to determine the relationship between Indigenous identity and campus healthcare utilization. Table 4 presents the logistic regression models for Campus Health (CH) use, showing the odds ratios (OR) and 95% confidence intervals (CI) for different models. These included the crude analysis, adjustments for the Perceived Stress Scale categorized score, for PSS score and class standing, and finally for PSS score, class standing, and a binomial gender variable. The gender variable was adjusted to only report those who reported male or female for this variable due to sparse numbers reported for other gender categories and protection of identity. The results showed that compared to the referent group (MEIM Tertile 1), the odds ratios (OR) for MEIM Tertile 2 and Tertile 3 were not statistically significantly different across all models. In these models, we observe that the 95% confidence intervals include one for every model, therefore, the true effect could range from a significant decrease to a substantial increase. For MEIM Tertile 2, the ORs ranged from 1.0 to 1.1, and for MEIM Tertile 3, the ORs ranged from 1.2 to 1.4. Similarly, the continuous MEIM score did not show statistically significant associations, with ORs ranging from 1.1 to 1.3 across the models. These findings suggest that MEIM, whether categorized or continuous, was not a strong predictor of Campus Health use in this sample.

Table 5 presents the logistic regression models for CAPS (Counseling and Psychological Services) use, showing the odds ratios (OR) and 95% confidence intervals (CI) for different models: the crude analysis, adjustments for the Perceived Stress Scale categorized score, for PSS score and class standing, and finally for PSS score, class standing, and a binomial gender variable. Like for Table 4, due to small numbers for the gender variable, only those who reported male or female for this variable are reported on. Like Table 4, the results for measuring the relationship between MEIM score and CAPS usage are not statistically different in models across any of the tertiles. For MEIM Tertile 2, the ORs ranged from 1.2 to 1.3, and for MEIM Tertile 3, the ORs ranged from 1.9 to 2.2. Similarly, the continuous MEIM score did not show statistically significant associations, with ORs ranging from 1.7 to 1.9 across the models. Like in Table 3, in these models, we observe that the 95% confidence intervals include one for every model, therefore, the true effect could range from a significant decrease to a substantial increase.

### 3.4. Healthcare Utilization and PSS Score 

The final part of this study is to observe the potential relationship between perceived levels of stress for Indigenous students and healthcare service utilization. Table 6 is showcasing PSS scores using the “low” PSS as the reference group compared to the “moderate” and “severe” scored groups. Given what was observed in Table 4 and Table 5, we examined a crude model of PSS score vs. CH/CAPS usage and then examined if being a graduate student and if gender were confounders. Odds ratios for campus health usage were 0.9 for moderate PSS scores and were between 0.2 and 0.6 for severe scores on the PSS. For CAPS usages, moderate scores ranged between 1.4 and 1.8 while scores were between 1.2 and 12.7 for severe scores on the PSS. We observed that for all models at all levels tested, the 95% confidence intervals included one, so the listed odds ratios cannot be concluded to be significant as the true value may be an increase or decrease.

### 3.5. Differences via Gender

Significant differences were observed in how students of different genders utilized Campus Health and CAPS. Students who identified as female were more 2.9 times more likely to utilize Campus Health services compared to students who identified as male (OR: 2.9, 95% CI: 1.1–7.9). Similar effects were seen with CAPS, but these were not significant.

### 3.6. Differences Vis Undergraduate vs. Graduate Students

We also observed differences in how undergraduate and graduate students reported utilization of on campus healthcare. Graduate students were far less likely to utilize CAPS (OR: 0.2, 95% CI: 0.07–0.5) in another statistically significant finding. There was a similar observation with Campus Health, but this was not significant.

## 4. Discussion

This study set out to look at the potential relationships between Indigenous identity among students, perceived stress of students, and the utilization of on campus healthcare among other factors. The first hypothesis heading into the study was that students who had a higher score on the MEIM would be less likely to utilize campus healthcare. Our second hypothesis is that students who scored in the higher categories for the PSS scale would be less likely to access CH and CAPS. The second hypothesis was that those who had higher scores on the MEIM would score lower on the PSS. While not statistically significant when measured continuously or in tertiles, we see that those that report a higher MEIM score are at higher odds of using campus health. We see a similar association with CAPS. Despite this, there are some interesting findings for future work. The first is that there appears to be a positive relationship between Indigenous identity and perceived stress. While not statistically significant, there is a trend showing some protective factors, which is in line with prior scoping review findings [10]. What may be complicated in this study is that almost all of our participants reported high scores on the identity measure.

### 4.1. Understanding Indigenous Identity’s Relationship with Campus Health Utilization 

In this study, we see that in sorting MEIM results into tertiles based on scoring, that a higher MEIM score among Indigenous students has no impact on if students are more or less likely to use Campus Health or CAPS. This is not entirely surprising, as the MEIM scores for almost all participants in this study were quite high; with the top two tertiles including participants who scored between 4.2 and 5.0 in the averages, which on the 5-point Likert scale is quite high. Our population scoring that high consistently on the identity measurements could mean two potential things. The first is that the university is providing adequate support for Native American students and help to foster their Indigenous identities. However, NASA only sees a fraction of the Native American identifying students attend their center and without further studies, we cannot verify this conjecture.

The second is that the instrument we are using for this study may have limitations with our specific population. In a prior scoping review from our research group, we showed that when using Indigenous identity to examine its relationship with stress, studies were split evenly between having a positive impact, negative impact, and no impact [10]. While this review focused on the relationship between Indigenous identity and stress, and this current study is homed in on the relationship between Indigenous identity and healthcare utilization, there are still connections to be made. Namely that the tools used in this study and the plurality of studies in the review are the same, the MEIM. While this is an effective tool for measuring identity with other racial and ethnic groups, there are challenges with using it with the Indigenous populations within the United States. Challenges such as the current MEIM and MEIM-R do not have questions around the enrollment status of Indigenous people, nor are there questions related to the Certified Degree of Indian Blood or multiracial status of many Indigenous people. Perhaps, with a better adapted scale or newly developed scale that includes questions not only on Indigenous peoples individual and cultural identities, but also one that factors in a person’s political status, or tribal enrollment status, as Indigenous identity is both self-perceived identity and possessing political rights [28], we may see more significant findings in future studies.

### 4.2. Cost as a Barrier to Care

While no relationship could be determined for the aim of this study, we did observe many potential additional significant relationships that should be further explored. The first was that we see a significant negative relationship between perceived affordability of healthcare and usage of CAPS. Students, on average, were 7.3 times less likely to use CAPS if they felt the cost was too expensive. Affordability of mental health care is continuously seen as a significant barrier to college students receiving care [29,30]. The perceived cost of healthcare may contribute to lower rates of utilization. If students expect the visit to be an expense they cannot afford at the present time, they may then choose to refrain from scheduling routine health appointments or utilizing urgent care services. Prior work with urban Indigenous populations has evaluated Urban Indian Organizations for how well they promote their programs to assist with cost of care, showing that work needs to be done to better showcase their range of services [31]. When applying this methodology to the CH website, we see that there is little said about programs to help manage costs of care and accessibility if the healthcare is out of network.

This may be a significant barrier for CAPS counseling as there is a co-pay for visits. It has been documented in prior research that mental health concerns can predict utilization of mental health services in the general student population [32]. So, while students may be interested, cost is a contributing factor. The university has attempted to combat this through the development of an embedded CAPS counselor in NASA, who is able to provide no-cost, drop-in visits for students, and this has been seen to improve usage of CAPS [33]. More work needs to be done to examine viability and feasibility of no-cost or low-cost options for CAPS and CH.

### 4.3. Insurance as a Barrier to Care

To restate, the university is seeing increased numbers of students from Indigenous communities attend the university while also seeing a decrease in the number of students who utilize Campus Health. Limited awareness and access to health insurance accepted at Campus Health may be a barrier. In the general population, nonelderly Indigenous people in the United States are more than twice as likely to be uninsured in comparison to White populations [34]. We see two interesting findings from this study that could be explored further. The first is that a considerable number of students report having AHCCCS. Medicaid has been vital for many IHS services to not just enroll more patients, but also to cover gaps in costs that IHS funding alone cannot [35]. While AHCCCS has made it so most Indigenous people are covered in the state, CH does not accept it at their locations, creating a barrier to care.

Likewise, this is reflected in the follow up question where we ask if students are required to go to a specific healthcare provider other than Campus Health for their insurance. While we see that the split is relatively even, with roughly 33% saying “Yes, I have to go elsewhere”, “No, I do not need to go elsewhere”, and “Unsure”, those who report saying “Yes” or “Unsure” were far less likely to use Campus Health (Yes OR: 0.2, 95% CI: 0.08–0.5); (Unsure OR: 0.09, 95% CI: 0.03–0.3) and CAPS (Yes OR: 0.2, 95% CI: 0.09–0.6); (Unsure OR: 0.2, 95%CI: 0.04–0.4). What this shows is that there may be a link between the cost of care, accepted insurances by CH, and usage rates and should be further explored. Past literature has identified a link between health insurance literacy and accessing health care among the general population and should also be explored with college students [36].

Although the university has one of the largest Native American student populations in the country, there are no questions in the Campus Healthcare utilization survey that specifically address Native American issues. The main two that can be pointed out are the questions above on insurance. It is well documented that IHS healthcare is not insurance [37], however, it would be important to ask in the insurance questions if students believe their primary insurance is “IHS”. This could shine a light on understanding the potential needs for more affordable options for routine and urgent care among Indigenous university students.

### 4.4. Give Back

As stated in the methods section, as part of acquiring letters of support for this project, deliverables, and final report out, meetings were scheduled after final analysis of this research concluded. The research team collaborated to design easy to digest printable materials with key findings in an appealing format for our partners to read in addition to the, or in place of, the manuscripts to be developed. Over the course of a month, the researchers met with CH, NASA, and NAATE, as well as the local Urban Indian Organization, Tucson Indian Center, to review findings, highlight barriers, and identify next steps. Campus partners then attended the PI’s community research presentations where findings were again highlighted and showcased to the community with hopes for future collaboration, such as a partnership between the college of public health and CH to provide patient and insurance navigation services to university students provided by public health students aiming to satisfy their work requirements. Work and following up continue in hopes that improvements to services for Indigenous students become a reality.

### 4.5. Limitations 

There are several limitations in this study. One of the primary limitations in this study is that the tools (survey) of CH were used, with no changes or modifications to questions or tailoring to be culturally responsive. As a result, certain choices relevant for Indigenous populations were not included in the final survey. For example, in none of the questions, we observe selections for “Indian Health Service” either by name or as an example of insurance. IHS is not insurance, but students may not be aware of this, and it may be good for follow up studies. This has been brought up to CH leadership as a direct result of this study and it has been recommended this be added to future utilization surveys going forward.

Second, while we did not see a relationship between Indigenous identity and healthcare utilization, this may be due to the sample size, demographics, or the instrument. Prior research by the PI has discussed the limitations of the MEIM for use with Indigenous populations in the United States as it does not touch on enrollment criteria and all aspects of Indigenous identity [28]. More research should be done to examine if there is a need for additional tools to be created specifically for an Indigenous identity measurement that incorporate Tribal Critical Race Theory [38].

A final limitation is that the University cites over 2000 “Native American” self-identifying students [1]. This is the number this study utilizes when determining sample size and power. The University’s current definition of “American Indian or Alaska Native” and “Native American” could be refined as they currently include Indigenous populations from outside the United States, including Canada, Mexico, or other parts of Central or South America. As this was an identified limitation of the data set/defining of the population, recommendations from this study were presented to partners and included more specificity in terminology when it comes to race/ethnicity classification for students and in public presentation-websites, etc. This can help lead to a precision of numbers that can be utilized to create tailored programming for university students as there are great differences among Indigenous peoples by tribes/Nations.

### 4.6. Future Directions

This study has shown that there needs to be further examination of the instruments that Campus Health and University admissions uses for their data collection to enhance data collection on the Indigenous student population, including the Native American subpopulation. The current campus health annual survey does not ask questions on if students utilize Indian Health Service. Follow-up studies should be done to examine the benefits of tailoring questions to target populations with lower rates of healthcare utilization both at our university and other universities with large populations of Indigenous identifying students. In addition, there remain concerns about the MEIM and its usefulness in measuring identity within Indigenous populations. As stated above, over two-thirds of participants in this survey identified as having strong Indigenous identities. While this may be true, there are concerns about if the MEIM asks questions that get at sources of conflict for Indigenous identity. The main contention is that while the MEIM and MEIM-R are the most widely used identity scales with Indigenous people, they do not have any questions around political identity, specifically around Tribal enrollment and blood quantum [28]. Future studies should be conducted to validate additional questions that ask about enrollment status, living away from community, and self and external perceptions from being in relationships with partners from outside their specific Tribal Nation.

## 5. Conclusions

While this study does not demonstrate statistical significance in the relationship between Indigenous identity, perceived stress, and healthcare utilization, there are many benefits for current and future healthcare utilization initiatives with Indigenous populations in the United States. There is confirmation that affordability of healthcare impacts Indigenous university students. Perceived cost of care is a statistically significant barrier to care for Indigenous students at the university. More work needs to be done to promote financial assistance and health literacy programs to aid Indigenous students in receiving quality health care.

## Figures and Tables

**Figure 1 ijerph-22-01409-f001:**
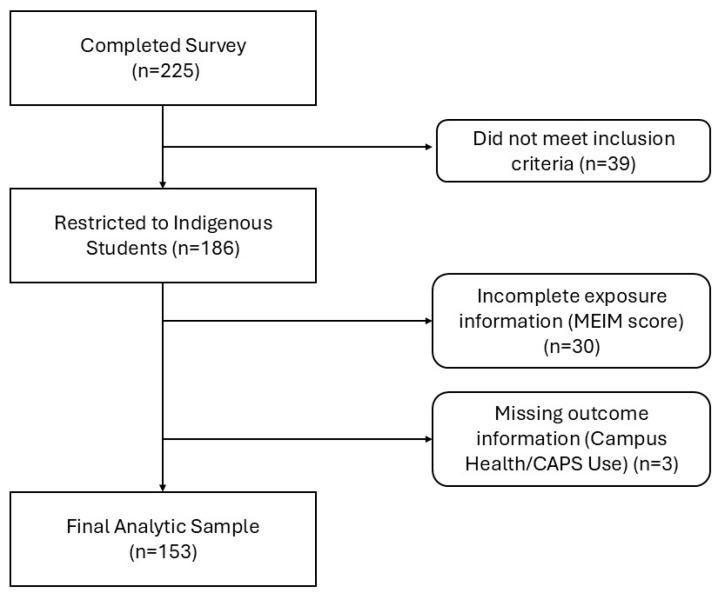
Inclusion/Exclusion Criteria.

**Table 1 ijerph-22-01409-t001:** Demographic Data for Makeup of Indigenous Student Population in Study.

	Total (N = 153)	MEIM Tertile 1 (2.1–4.1) (n = 52)	MEIM Tertile 2 (4.2–4.6) (n = 46)	MEIM Tertile 3 (4.7–5.0) (n = 55)
Gender				
Male	31 (20.3)	12 (23.1)	9 (19.6)	10 (18.2)
Female	118 (77.1)	38 (73.1)	36 (78.3)	44 (80.0)
Another Gender Identity	4 (2.6)	2 (3.9)	1 (2.2)	1 (1.8)
NA/AN and one or more races?	42 (27.5)	22 (42.3)	11 (23.9)	9 (16.4)
Class Standing				
Undergraduate	107 (69.9)	40 (76.9)	33 (71.7)	34 (61.8)
Graduate	45 (29.4)	12 (23.1)	13 (28.3)	20 (36.4)
Missing	1 (0.7)			
Enrolled Member of Tribal Nation				
Yes	142 (92.8)	46 (88.5)	43 (93.5)	53 (96.4)
No	10 (6.5)	5 (9.6)	3 (6.5)	2 (3.6)
Missing	1 (0.7)	1 (1.9)	-	-
Number of Tribal Affiliations				
One	128 (83.7)	44 (84.6)	34 (73.9)	50 (90.9)
≥2	25 (16.3)	8 (15.4)	12 (26.1)	5 (9.1)
Perceived Stress				
Low	28 (18.3)	8 (15.4)	10 (21.7)	10 (18.2)
Moderate	111 (72.6)	39 (75.0)	33 (71.7)	39 (70.9)
Severe	14 (9.2)	5 (9.6)	3 (6.5)	6 (10.9)

**Table 2 ijerph-22-01409-t002:** Reporting on Healthcare considerations, Insurance, and other instruments.

	Total (N = 153)
Consider Campus Health/CAPS	
Yes	94 (61.4)
No	59 (38.6)
Type Of Health Insurance	
College/University Sponsored Plan	22 (14.4)
Parent Plan	67 (43.8)
ACCHS	25 (16.3)
Another Type of Plan	22 (14.4)
No Health Insurance	16 (10.5)
Unsure	1 (0.6)
Insurance Elsewhere	
Yes	48 (31.4)
No	50 (32.7)
Unsure	54 (35.3)
Missing	1 (0.7)

**Table 3 ijerph-22-01409-t003:** Odds Ratios for Perceived Healthcare Costs vs. Utilization of Services.

Does Your Insurance Require You to Go Elsewhere for Healthcare?	Odds Ratio	95% Confidence Interval	
Yes	0.2	0.08	0.5
Unsure	0.09	0.03	0.3
Does Your Insurance Require You to Go Elsewhere for Psychological Services?	Odds Ratio	95% Confidence Interval	
Yes	0.2	0.09	0.5
Unsure	0.2	0.04	0.4

**Table 4 ijerph-22-01409-t004:** Logistic Regression Models for MEIM Score vs. Campus Health Use.

	Model 1	Model 2	Model 3	Model 4
	OR (95% CI)	OR (95% CI)	OR (95% CI)	OR (95% CI)
MEIM (Categorized)				
MEIM Tertile 1	Referent	Referent	Referent	Referent
MEIM Tertile 2	1.1 (0.5, 2.6)	1.0 (0.4, 2.5)	1.1 (0.4, 2.7)	1.1 (0.4, 2.7)
MEIM Tertile 3	1.4 (0.6, 3.2)	1.2 (0.5, 2.8)	1.4 (0.6, 3.1)	1.4 (0.5, 3.3)
MEIM Score (Continuous)	1.3 (0.7, 2.4)	1.3 (0.7, 2.1)	1.1 (0.6, 2.2)	1.2 (0.6, 2.5)
Model 1: Crude Model 2: Adjusted for PSS Score Model 3: Adjusted for PSS Score, Class Standing, Model 4: Adjusted for PSS Score, Class Standing, Gender (Male vs. Female)				

**Table 5 ijerph-22-01409-t005:** Logistic Regression for MEIM Score vs. CAPS Use.

	Model 1	Model 2	Model 3	Model 4
	OR (95% CI)	OR (95% CI)	OR (95% CI)	OR (95% CI)
MEIM (Categorized)				
MEIM Tertile 1	Referent	Referent	Referent	Referent
MEIM Tertile 2	1.2 (0.5, 3.0)	1.2 (0.5, 3.1)	1.3 (0.5, 3.6)	1.2 (0.4, 3.7)
MEIM Tertile 3	2.0 (0.8, 4.7)	2.0 (0.8, 4.8)	2.2 (0.9, 5.6)	1.9 (0.7, 5.4)
MEIM Score (Continuous)	1.7 (0.8, 3.5)	1.7 (0.8, 3.5)	1.9 (0.9, 4.2)	1.7 (0.7, 3.9)
Model 1: Crude Model 2: Adjusted for PSS Score Model 3: Adjusted for PSS Score, Class Standing, Model 4: Adjusted for PSS Score, Class Standing, Gender (Male vs. Female)				

**Table 6 ijerph-22-01409-t006:** Logistic Regression for PSS Score vs. Campus Health/CAPS Use.

	Campus Health Usage	CAPS Usage
Perceived Stress Score	OR (95% CI)	OR (95% CI)
Model 1		
Low (PSS < 13)	Referent	Referent
Moderate (PSS ≤ 13 and ≥26)	0.9 (0.4, 2.1)	1.4 (0.5, 3.8)
Severe (PSS > 26)	0.5 (0.1, 2.2)	1.5 (0.3, 6.4)
Model 2		
Low (PSS < 13)	Referent	Referent
Moderate (PSS ≤ 13 and ≥26)	0.9 (0.4, 2.3)	1.8 (0.6, 5.5)
Severe (PSS > 26)	0.6 (0.1, 2.9)	12.7(0.5, 13.6)
Model 3		
Low (PSS < 13)	Referent	Referent
Moderate (PSS ≤ 13 and ≥26)	0.9 (0.4, 2.3)	1.7 (0.5, 5.3)
Severe (PSS > 26)	0.2 (0.02, 1.8)	1.2 (0.2, 8.2)
Model 1: Crude Model 2: Adjusted Class Standing, Model 3: Adjusted Class Standing, Gender (Male vs. Female)		

## Data Availability

The data presented in this study are available on request from the corresponding author on a case by case basis to respect to the principles of Indigenous Data Sovereignty as data is generated from Indigenous populations.

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
