# Peer review of "Evaluating Indigenous Identity and Stress as Potential Barriers to Accessing On-Campus Healthcare for Indigenous Students at a Large Southwestern University"

_ijerph, 2025, doi:10.3390/ijerph22091409_

Round 1

Reviewer 1 Report

Comments and Suggestions for Authors

Significance- This study addresses the urgent issue of underutilization of campus healthcare services among Indigenous college students. The potential for concrete and tangible impacts on campus healthcare systems to improve utilization rates among Indigenous students, a population often experiencing significant health inequities, is clear and compelling.

Introduction-The primary weakness of this study lies in the lack of a clear and compelling rationale for its purpose. For instance, the Introduction section does not comprehensively or clearly provide and cite appropriate literature explaining how or why Indigenous identity might specifically influence healthcare utilization. While some studies on the relationship between Indigenous identity and stress are described, the authors fail to make a clear case for examining Indigenous identity and stress in relation to healthcare utilization. It is unsurprising, then, that the study findings revealed these two factors were not statistically significantly related to healthcare utilization. Instead, systemic factors, such as insurance and affordability, were found to be influential. It also remains unclear why these two significant factors were not included in the model as confounders. Similarly, the question regarding traditional healers is mentioned in the Methods section, but its inclusion in the analysis, if any, is not elucidated.

Methods-The manuscript mentions TribalCrit as the theoretical framework for this study, yet the authors do not clearly articulate its application to the study design, nor is it referenced again beyond a single paragraph. There is no evidence of how TribalCrit informed any phase of this study. Therefore, I strongly recommend removing this paragraph from the manuscript.

It is also unclear whether the Multi-Ethnic Identity Measure (MEIM) is an appropriate measure of Indigenous identity for this context. The authors appear to inconsistently argue for and against its use throughout the manuscript. Furthermore, the rationale for categorizing the composite scores for MEIM into tertiles is not clear.

Several topics are mentioned in the manuscript without adequate context. For example, it doesn't become apparent until later in the manuscript (line 174) that this study was part of a dissertation. Indigenous Data Sovereignty is also mentioned but not explained. Additionally, compensation is discussed in the Results section but not described in the Methods.

Results-The Results section would greatly benefit from clearer organization. For example, the tables could be better structured and made more concise. The perceived stress variable, for instance, could be moved to Table 1 to better display the MEIM tertiles. It is also confusing why "considering using Campus Health/CAPS" is presented as a variable instead of actual utilization in Table 2. The paragraphs discussing differences in gender and class standing should be moved to appear before the model descriptions. Finally, as noted previously, it is not clear why insurance status and affordability were not included in the models.

Discussion-The study hypotheses are mentioned for the first time in the Discussion section, creating a disjointed narrative from the studies reviewed in the Introduction. For example, it is not clear why the authors would hypothesize that higher MEIM and stress scores would lead to less utilization. These hypotheses should be introduced and supported with cited studies in the Introduction. The high MEIM scores of the sample are also mentioned for the first time in this section. The hypothesis that this reflects the University of Arizona doing an "exceptional job" of supporting these students needs to be better substantiated with evidence. Talking Circles are also introduced for the first time without context or citation. It would be valuable to learn more about this and how Talking Circles could play a role in the future direction of this study. Given that identity is a complex construct, qualitative research could provide more comprehensive insights into this issue.

Broader Issue-A more fundamental issue with this study is its weak and unclear justification of the research question. A growing body of research demonstrates that systemic inequities are largely responsible for healthcare disparities. Focusing too closely on individual-level factors, such as identity, can inadvertently perpetuate a victim-blaming narrative. This study would be significantly more impactful if its framing were adjusted to center the statistically significant relationship between insurance and affordability and healthcare utilization, aligning with a more systemic understanding of health disparities.

The manuscript requires a thorough review for incomplete sentences and syntax errors.

Author Response

Significance- This study addresses the urgent issue of underutilization of campus healthcare services among Indigenous college students. The potential for concrete and tangible impacts on campus healthcare systems to improve utilization rates among Indigenous students, a population often experiencing significant health inequities, is clear and compelling.

Introduction-The primary weakness of this study lies in the lack of a clear and compelling rationale for its purpose. For instance, the Introduction section does not comprehensively or clearly provide and cite appropriate literature explaining how or why Indigenous identity might specifically influence healthcare utilization. While some studies on the relationship between Indigenous identity and stress are described, the authors fail to make a clear case for examining Indigenous identity and stress in relation to healthcare utilization. It is unsurprising, then, that the study findings revealed these two factors were not statistically significantly related to healthcare utilization. Instead, systemic factors, such as insurance and affordability, were found to be influential. It also remains unclear why these two significant factors were not included in the model as confounders. Similarly, the question regarding traditional healers is mentioned in the Methods section, but its inclusion in the analysis, if any, is not elucidated.

  • These are all highly relevant and important points raised, we thank you for this.
  • We thank you for asking to further clarify the relationship between Indigenous identity and perceived stress to healthcare utilization and how it came to be used in this study. This came from a community based approach through working with our campus partners. As noted in the paper, on line 93-94, Indigenous students are reporting the lowest access levels to healthcare on campus and the assumption by healthcare and faculty is that there is a potential cultural barrier; that students actively choose to go home for care instead of use care on campus. This is the premise of the study and we have added additional clarifications here as well as a formal hypothesis. See more on lines 96-101. Our hypothesis was meant to see if there is an explanation grounded in cultural differences that explain why Indigenous students use healthcare. This would require a great investment of both resources and infrastructure to bridge. Instead we see that, to the best of knowledge, no such barriers exist for this university and it is instead similar factors to the larger population. This was a profound result which all community partners were both surprised and energized by. The null hypothesis failing to be rejected in this case was just as valuable to see as a validation of the hypothesis. As stated in the “give back” section of the paper, work is already being done to increase community outreach.
  • We removed mention of the question on Traditional Healers, as while it was in the survey, the data did not make it to this manuscript. The questions asked about if students would be interested in access to a traditional healer and a follow up asked if they would use a traditional healer from outside their tribe. When asked generally, students overwhelmingly answered “yes” they would like a healer. When asked if they would access a traditional healer from another tribe, students answered with an overwhelming “no”. This was another key finding for the community partners as one of the ways in which universities may seek to be more involved in Indigenous cultures.
  • We are added information on lines 94-96 showing the potential ways in which ethnic identity can be linked to healthcare utilization
  • Finally, we understand the question raised about the confounders and we did run statistical tests. The issue is that they also did not show significant results , and with this, we decided to stick with the original data sets. The real issue we observed is that the Identity scores with the MEIM are all so high that there is no differentiating between groups.

Methods-The manuscript mentions TribalCrit as the theoretical framework for this study, yet the authors do not clearly articulate its application to the study design, nor is it referenced again beyond a single paragraph. There is no evidence of how TribalCrit informed any phase of this study. Therefore, I strongly recommend removing this paragraph from the manuscript.

  • Thank you for this comment, after reviewing the manuscript again, we tend to agree as while the TribalCrit is an important theory for our group, it is not necessarily impacting how this portion of the dissertation was designed. We have removed mention of it in the methods.

It is also unclear whether the Multi-Ethnic Identity Measure (MEIM) is an appropriate measure of Indigenous identity for this context. The authors appear to inconsistently argue for and against its use throughout the manuscript. Furthermore, the rationale for categorizing the composite scores for MEIM into tertiles is not clear.

  • Thank you for this comment. Yes, we do maintain issue with the MEIM for studies with Indigenous populations. This comes from a past study done by this research team, a scoping review seen here, https://www.mdpi.com/1660-4601/21/11/1404. In this we see there are multiple conflicting studies which utilize Indigenous identity and stress and in our manuscript, we deem all tools to measure identity to be flawed given the lack of attention to questions around tribal enrollment and blood quantum. Therefore, we used the MEIM as it is the most frequently utilized tool for measuring Indigenous identity.
  • Per your comment on tertiles, this is very informative, and we chose to do this because the official guidance for the MEIM does not include anything but a continuous score. We chose to also do tertiles because this allowed us to create categories with reasonable distributions and sample sizes to see if there was any relationship between a “higher” Indigenous identity vs a low score. This has been prior done in other studies with Indigenous populations, see line 211-212 for sources.

Several topics are mentioned in the manuscript without adequate context. For example, it doesn't become apparent until later in the manuscript (line 174) that this study was part of a dissertation. Indigenous Data Sovereignty is also mentioned but not explained. Additionally, compensation is discussed in the Results section but not described in the Methods.

  • Thank you for these additional comments. We have removed mention of the dissertation on lines 182 and 486. As for Indigenous Data Sovereignty, it is mentioned because all data collected on Tribal Nations is Tribal data according to IDSov. If the research had collected data on the specific Tribal Nation affiliations of participants, we would be obligated to consult with each Tribe to obtain approval/consent from each Tribal Nation to move forward with the study and publication. Finally, we updated information on the compensation on 209-211.

Results-The Results section would greatly benefit from clearer organization. For example, the tables could be better structured and made more concise. The perceived stress variable, for instance, could be moved to Table 1 to better display the MEIM tertiles. It is also confusing why "considering using Campus Health/CAPS" is presented as a variable instead of actual utilization in Table 2. The paragraphs discussing differences in gender and class standing should be moved to appear before the model descriptions. Finally, as noted previously, it is not clear why insurance status and affordability were not included in the models.

  • For the question on why we refer to “considering using Campus Health/CAPS”, this is because it is the direct language from the survey questionnaire used by Campus Health.
  • You make an excellent point about the models. We did not include insurance status and affordability in the final models as they do not change the results. We attempted several different models and the issue lies with the MEIM scores, not with anything else. So, we chose to use our initial models and then include the insurance and affordability results as a separate table afterwards.

Discussion-The study hypotheses are mentioned for the first time in the Discussion section, creating a disjointed narrative from the studies reviewed in the Introduction. For example, it is not clear why the authors would hypothesize that higher MEIM and stress scores would lead to less utilization. These hypotheses should be introduced and supported with cited studies in the Introduction. The high MEIM scores of the sample are also mentioned for the first time in this section. The hypothesis that this reflects the University of Arizona doing an "exceptional job" of supporting these students needs to be better substantiated with evidence. Talking Circles are also introduced for the first time without context or citation. It would be valuable to learn more about this and how Talking Circles could play a role in the future direction of this study. Given that identity is a complex construct, qualitative research could provide more comprehensive insights into this issue.

  • Thank you for this comment. We have included the hypothesis on lines 96-101.
  • Thank you for pointing out the comment about “exceptional” on line 401. This is not meant to be a takeaway of ours, but instead a way to read the MEIM results. We have also added in direct comments on the tertiles in lines 280-291.
  • We agree that qualitative research could be a benefit (we have a qualitative piece that will be published as a stand along manuscript) but we also point to prior work done by the researchers which shows there is a great deal of research on measuring Indigenous identity in quantitative work. Given the results of this study, and other recent manuscripts in the prior scoping review, we believe more future work is needed to ensure quantitative measurement tools are incorporating results from qualitative research and Indigenous theories and frameworks.

Broader Issue-A more fundamental issue with this study is its weak and unclear justification of the research question. A growing body of research demonstrates that systemic inequities are largely responsible for healthcare disparities. Focusing too closely on individual-level factors, such as identity, can inadvertently perpetuate a victim-blaming narrative. This study would be significantly more impactful if its framing were adjusted to center the statistically significant relationship between insurance and affordability and healthcare utilization, aligning with a more systemic understanding of health disparities.

  • We have updated the paper to better emphasize the community-based approach of the study. The intention was always to challenge the held assumptions of the university community and given what is discussed prior to the implementation of research, and in the literature, this was an appropriate subject. We only included the traditional measures of health disparities after being unable to reject the null hypothesis as that would also be of interest to our partners. We also challenge that Indigenous identity is too much of an individual level factor. As we have shown in prior work, Indigenous identity is well researched, and ethnic identity is well-studied in health sciences as well, with it being understood as a potential buffer to adverse psychosocial events. A strong ethnic identity can assist in both accessing treatment and the effectiveness of interventions. The social and political determinants of health, of which belonging, acceptance, and legal enrollment in a Tribal Nation is part of, are well documented.

The manuscript requires a thorough review for incomplete sentences and syntax errors.

Reviewer 2 Report

Comments and Suggestions for Authors

This is a well and clearly written paper investigating an interesting and important question of access to care for Indigenous students at one US university. Given the single site design and the low participation rate in the survey on which the analysis is based, there are some significant limitations to the lessons to be learned from this study, and the authors cannot change this. Recognizing that, there are also interesting observations and areas for future work highlighted that make the paper of interest. The finding that many students had uncertainty about whether their insurance covered campus care has important practical implications.

Specific suggestions for revisions:

  • The paper needs to acknowledge more directly the limitations of what is likely a non-representative sample. Only 157 out of a potential 2000, or less than 10%, Indigenous students at the university participated in the study. This is not just about small numbers; the students who chose to complete the survey might differ in some important way from those who did not. The paper mentions that students needed to provide a university email to receive compensation - what was the compensation? Might this incentive, if small (as it probably was, consistent with common practice and budget constraints), made it more likely that the sample includes more student under greater financial stress? These issues in sampling, both size and potential non-representativeness, should be better acknowledged in the limitations, I think.
  •  In Discussion section 4.2 we learn there is a co-pay for mental health services. I suggest stating this earlier in the background section, as well as if there is a co-pay for primary care services, as I was wondering this already and it is an important contextual factor to know.
  • The paper reports that "MEIM scores for almost all participants in this study were quite high" (line 375). Is there any literature to cite to figure out whether this is a frequent outcome with the MEIM when used with this population i.e., it may not be capturing quite the granularity that would be useful for studies of Indigenous populations in the US? The authors suggest this explanation around lines 390-393. An alternative explanation not considered comes back to the sample: is this possibly one of those ways in which the study sample might differ importantly from the larger Indigenous student body? If participants were recruited often by word of mouth through events hosted by NASA, for example, these students are likely to feel more connected to their Native identity. 

I think the discussion raises many thoughtful points and is adequately comprehensive. One thing I would suggest for the authors' consideration, whether they wish to work it into the paper or not, relates to the paragraph already mentioned above (lines 385-401) and the role of context in how identity matters. The authors suggest that the MEIM may not be that helpful for understanding/predicting stress in this population because it does not identify/measure all the complexities of Indigenous identity. I would offer that the challenge in finding a consistent relationship between (Indigenous) identity and stress may have its origins more. or at least equally, in context and not in individual identity, however well measured. In some settings and scenarios, it seems plausible that stronger Native identity will be protective against stress, while in other contexts and settings it may exacerbate stress.   

Finally, just as appreciations: I liked that the authors included information about a bot attack and the challenges this created - this is a "new normal" that it is helpful to see named. I also appreciated the inclusion of the "give back" section and the subtle but important references to the ways in which the study design considered and incorporated Indigenous Data Sovereignty principles. 

Author Response

This is a well and clearly written paper investigating an interesting and important question of access to care for Indigenous students at one US university. Given the single site design and the low participation rate in the survey on which the analysis is based, there are some significant limitations to the lessons to be learned from this study, and the authors cannot change this. Recognizing that, there are also interesting observations and areas for future work highlighted that make the paper of interest. The finding that many students had uncertainty about whether their insurance covered campus care has important practical implications.

Specific suggestions for revisions:

  • The paper needs to acknowledge more directly the limitations of what is likely a non-representative sample. Only 157 out of a potential 2000, or less than 10%, Indigenous students at the university participated in the study. This is not just about small numbers; the students who chose to complete the survey might differ in some important way from those who did not. The paper mentions that students needed to provide a university email to receive compensation - what was the compensation? Might this incentive, if small (as it probably was, consistent with common practice and budget constraints), made it more likely that the sample includes more student under greater financial stress? These issues in sampling, both size and potential non-representativeness, should be better acknowledged in the limitations, I think.
    • Thank you for this insightful comment. We appreciate it and have made additional comments in our limitations, as seen on lines 496-506. This goes into the way “Native American” students were defined by our partners. We also point back to our power analysis done on 199-206 which gives us appropriate power for this study with 153 participants.
  •  In Discussion section 4.2 we learn there is a co-pay for mental health services. I suggest stating this earlier in the background section, as well as if there is a co-pay for primary care services, as I was wondering this already and it is an important contextual factor to know.
    • This has been acknowledged and updated in lines 84-85 in the introduction. Thank you for catching this.
  • The paper reports that "MEIM scores for almost all participants in this study were quite high" (line 375). Is there any literature to cite to figure out whether this is a frequent outcome with the MEIM when used with this population i.e., it may not be capturing quite the granularity that would be useful for studies of Indigenous populations in the US? The authors suggest this explanation around lines 390-393. An alternative explanation not considered comes back to the sample: is this possibly one of those ways in which the study sample might differ importantly from the larger Indigenous student body? If participants were recruited often by word of mouth through events hosted by NASA, for example, these students are likely to feel more connected to their Native identity. 
    • Thank you, this is of interest to us. We do acknowledge that this may be a consideration but we also believe that due to our added in third limitation, and the ways in which our study was able to recruit, with a huge variety of methods in person, at events, and over the internet, that we perceive it as an appropriate sample. This paper emerged from a dissertation and findings from a qualitative study not yet published showing that even Indigenous students who go to NASA question their identities in a strong manner. There appears to be something missing in our identity measurement when struggles with Indigenous identity is discussed by our community partners and in our qualitative research but not reflected in our quantitative data, as more than 75% of participants scored highly.

I think the discussion raises many thoughtful points and is adequately comprehensive. One thing I would suggest for the authors' consideration, whether they wish to work it into the paper or not, relates to the paragraph already mentioned above (lines 385-401) and the role of context in how identity matters. The authors suggest that the MEIM may not be that helpful for understanding/predicting stress in this population because it does not identify/measure all the complexities of Indigenous identity. I would offer that the challenge in finding a consistent relationship between (Indigenous) identity and stress may have its origins more. or at least equally, in context and not in individual identity, however well measured. In some settings and scenarios, it seems plausible that stronger Native identity will be protective against stress, while in other contexts and settings it may exacerbate stress.   

  • Thank you for these comments again, and we understand this. The research team has conducted a scoping review on this subject, previously published (and cited in the manuscript, citation #10), which highlighted issues with how identity is measured and in which situations identity can be beneficial, but also hindering or no affect at all. Please see the following link: https://www.mdpi.com/1660-4601/21/11/1404. We also want to point to a recent article that also used the used the MEIM to measure identity for academic outcomes and found identity not impacting either, for similar reasons to our study, https://doi.org/10.1037/dhe0000494. To reiterate, we understand that identity is complex and Indigenous identity even more so, due to unique challenges each colonizing country has imposed on the Indigenous people in the occupied territory. For the US, it is of interest to us that our main tools to measure Indigenous identity do not include questions on enrollment status, which is a major issue in AI/AN communities.

Finally, just as appreciations: I liked that the authors included information about a bot attack and the challenges this created - this is a "new normal" that it is helpful to see named. I also appreciated the inclusion of the "give back" section and the subtle but important references to the ways in which the study design considered and incorporated Indigenous Data Sovereignty principles. 

  • Thank you for this. Yes, the bot attack was a major learning experience for us. In addition, the “Give Back” was crucial as the data from this study needed to be shared out with our partners prior to any other work being done.